# OpenReview forum: "Cancer-Myth: Evaluating Large Language Models on Patient Questions with False Presuppositions"
_ICLR.cc/2026/Conference — ICLR 2026 Poster_

### Official Review · Reviewer_1HfB · 2025-10-31

**Soundness:** 2
**Presentation:** 2
**Contribution:** 2
**Rating:** 4
**Confidence:** 4

**Summary:**

The paper tackles a blind spot in medical LLM evaluation: models often sound helpful yet miss or fail to correct false presuppositions in patient questions, which can mislead care. The authors compile 994 cancer myths and build Cancer-Myth, an expert-validated adversarial set of 585 patient-style questions with embedded misconceptions, plus Cancer-Myth-NFP (150 questions without false presuppositions) to gauge over-caution. They introduce a generate–respond–verify pipeline, followed by hematology–oncology physician review and category labeling. Evaluation is standardized with two metrics, PCS and PCR, that score whether answers fully correct the misconception. Using this setup, they show that frontier models and even a strong multi-agent system still struggle here, underscoring presupposition handling as a distinct skill from general medical QA.

**Strengths:**

- Strong, clinically grounded motivation. The paper highlights a practical and consequential gap in medical reasoning, illustrated with a clear example.
- Credible data contribution. The dataset appears carefully curated, and the generation–response–verification pipeline is transparent and methodologically sound.

**Weaknesses:**

- Missing frontier “reasoning-first” baselines with scaled test-time compute. The paper doesn’t evaluate models like OpenAI o1 or DeepSeek-R1, which are explicitly designed to improve as you give them more “think time” (test-time compute) and often change error profiles on reasoning tasks. At minimum, include these (or close open-source proxies) and report PCS/PCR vs. think-time (e.g., few-shot/none, best-of-N, majority vote), so readers can see whether presupposition correction improves with deliberate inference.
- “Expert-in-the-loop” needs clearer, auditable detail. Spell out the clinical review protocol: expert count and specialties; inclusion/exclusion criteria; annotation guidelines; whether experts had references; inter-rater reliability and adjudication; examples of borderline cases; and any pre/post expert edits to generated questions. Packaging this as a short datasheet/data statement (motivation, composition, collection, labeling, uses, maintenance) would make the dataset easier to trust and reuse.

**Questions:**

see weakness part

---

> ### Author Response · Authors · 2025-11-20
>
> We thank the reviewer for recognizing the strong, clinically grounded motivation of our work and evaluating our data contribution as credible, carefully curated, and transparent. We appreciate the constructive feedback and address your specific questions below.
>
> > W1: Missing frontier “reasoning-first” baselines with scaled test-time compute. The paper doesn’t evaluate models like OpenAI o1 or DeepSeek-R1, which are explicitly designed to improve as you give them more “think time” (test-time compute) and often change error profiles on reasoning tasks.
>
> We respectfully clarify that our evaluation suite includes GPT-5, Gemini-2.5-Pro, and Claude-4-Sonnet, which represent the current frontier of reasoning-enhanced architectures. Furthermore, DeepSeek-R1 was included in our paper (Figure 5), achieving a low PCR of 13.7%.
>
> Following the suggestion, we conducted additional experiments with more reasoning models OpenAI’s o3 and o4-mini, recording their average thinking time per question. As shown in the table below, increased inference compute does not solve the problem. The results suggest that while test-time compute improves complex logic (e.g., math, coding), it **does not inherently mitigate sycophancy or the tendency to accept false premises** in a user's prompt.
>
> Models|PCR|Thinking time per question
> :---|:---: |:---:
> GPT-5|42.1|24.5
> Gemini-2.5-Pro|41.4|33.6
> Claude-4-Sonnet|40.0|6.5
> O3|34.8|28.0
> DeepSeek-R1|13.7|52.4
> O4-mini|13.0|14.1
>
> Regarding sampling strategies (e.g., Best-of-N, majority votes), we excluded them to align with the real-world clinical deployment scenario, where patients interact with chatbots in a single-turn or conversational manner and do not aggregate multiple samples to derive a truth. We note that **we did test an agentic setup (MDAgents)**, which also failed to robustly correct misconceptions.
>
> > W2: “Expert-in-the-loop” needs clearer, auditable detail. Spell out the clinical review protocol.
>
> We appreciate the suggestion to improve auditability. We respectfully point out that the core expert protocol and quantitative metrics are currently documented in the **main text (Lines 293–300) and Appendix C.1.3**. Specifically, the paper currently details:
>  - Expert count and specialties (L293-L300): Three hematology oncology physicians;
>  - Inclusion/exclusion criteria (L293-L300): Experts applied two criteria **(1)** the generated patient question contains a false presupposition; **(2)** the factual correction is medically sound and appropriately addresses the false presupposition.
>    - Both (1) and (2) pass –> Cancer-Myth;
>    - Fail on (1) –> Cancer-Myth-NFP;
>    - Others –> Discarded.
>  - Inter-rater agreement (L293-L300): 83% of examples had full agreement across all three physicians, with a Gwet’s AC1 of 0.78, indicating substantial agreement.
>  - Borderline Cases (Appendix C.1.3): Qualitative analysis of disagreements is provided in the appendix.
>
> To fully address your request for a transparent audit trail, we will **add a Data Statement in the revision that includes the specific annotation guidelines** provided to the experts, as shown below:
>  - Annotation guidelines:
> > Dear Dr. XXX,
> >
> >  > Our previous cancer-care-related annotations and interviews highlighted that while AI chatbots perform well on many questions, they can sometimes provide harmful or incorrect answers, particularly when the questions contain false presuppositions or myths. Over the past two months, we have compiled a cancer myth dataset focused on patient questions that include false presuppositions.
> >
> > > As a reminder, our goal is to assess AI chatbots' ability to identify and correct medical myths in patient inquiries. Since these myths were sourced from various websites and the questions were generated using language models, we need your expertise to verify:
> >
> > > (1) Whether the question (Column A) contains a myth or false presupposition.
> >
> > > (2) Whether the provided information (Column B) is medically sound and appropriately addresses the false presupposition.
> >
> > > Each of you has been assigned ~300 examples here, [XXX’s Link]. The task should take approximately 3-4 hours.
> >
> > > Your primary focus is annotating Column C; if either (1) or (2) fails, the annotation should be marked as “No”, otherwise “Yes”. Please provide comments in Column D on where and why it fails if Column C is “No”.
>  - Pre/post expert edits to generated questions: no
>  - Whether experts had references: no
>
>
> > Flag For Ethics Review: Potentially harmful insights, methodologies and applications
>
> We are unable to find specific details regarding the concern in the review text. Could you please point us to the relevant section or clarify the specific grounds for the flag so that we may address it?

---

### Official Review · Reviewer_ct9K · 2025-10-31

**Soundness:** 3
**Presentation:** 3
**Contribution:** 4
**Rating:** 10
**Confidence:** 5

**Summary:**

This paper studies how well LLMs handle realistic cancer patient questions when those questions contain *false presuppositions* (e.g.  that there is no treatment for advanced lymphoma. The authors show current LLMs often give medically correct information but fail to challenge the false presupposition, which can have the effect of reinforcing the use of such presuppositions in question-answering. They build an expert-verified adversarial dataset to measure this and show that naïve “be cautious” prompting fixes one side of the problem but breaks the other.

Contributions

* Identifies and formalizes detecting and addressing false-presuppositions as a missing capability in medical LLMs for oncology patient Q&A (beyond MedQA-style benchmarks).
* Creates a Cancer-Myth dataset with 585 expert-verified, adversarial, cancer-specific patient questions that embed common cancer myths generated across 7 categories and validated by hematology–oncology physicians.
* Creates a companion Cancer-Myth-NFP set of 150 cancer questions with no false presuppositions for a baseline
* Shows that cutting-edge LLMs fail to fully correct the false presupposition much of the time, and that agent models also fail.
* Precautionary prompting can push correction on Cancer-Myth up but causes (i) a high rate of false positives on Cancer-Myth-NFP and (ii) a relative drop on standard medical benchmarks. This indicates that prompt engineering alone won't fix the problem.
* Reveals a clinically relevant tradeoff—improving myth-detection vs. preserving accuracy on valid patient questions.

**Strengths:**

Originality

* Frames false presuppositions as an evaluation target—this contributes to a broader set of work that seeks to evaluate beyond simple accuracy to addressing medical bias and logical fallacies that could be reinforced by LLMs and thus cause harm.
* But unlike some of that previous work, it provides an important medical benchmark.
* The choice to have the companion dataset that acts as a benchmark and helps with possibility of over-detection of false presuppositions is non-obvious and is a good touch.

Quality

* The dataset construction pipeline is careful: start from 994 real cancer myths, to LLM generation, to adversarial filtering, to physician verification. The  human-in-the-loop step makes the benchmark more credible for clinical safety use.
* The evaluation is applies to a broad set of models including agents.

Clarity

* The failure mode is illustrated with a simple, memorable example (advanced lymphoma → “no treatment”) and the paper keeps returning to that pattern, so the reader always knows what “false presupposition” means in context.

Significance

* The problem is clinically meaningful: reinforcing a false “no treatment” belief is more dangerous than giving a slightly incomplete chemo explanation.
* Addresses plausible real-world harm.
* The result that aggressive safety prompting boosts Cancer-Myth result but hurts Cancer-Myth-NFP result highlights an important trade-off
* Framework applies way beyond oncology and medics

**Weaknesses:**

* Relied on zero-shot prompting, doesn't explore best practice prompting techniques. Including prompts that included exemplars with facts might have impacted whether or not models correct presuppositions.
* No analysis of the reasons for the asymmetries of presupposition performance across categories.

**Questions:**

* How was the taxonomy for the seven myth categories derived?
* Can you provide any explanation or additional analysis that gives insight into why performance differs across myth categories?

---

> ### Author Response · Authors · 2025-11-20
>
> We thank the reviewer for their comprehensive summary and for recognizing the clinical significance of our work. We appreciate your assessment that our problem framing, dataset construction, and evaluation design constitute a strong and original contribution to the field of medical LLM safety. We address your specific questions and comments below.
>
> > W1: Relied on zero-shot prompting, doesn't explore best practice prompting techniques. Including prompts that included exemplars with facts might have impacted whether or not models correct presuppositions.
>
> We deliberately prioritized zero-shot prompting to **align with the most common real-world deployment scenario**: patients interacting with medical chatbots directly. In these “in-the-wild” settings, the system cannot anticipate the specific medical condition or myth a patient might introduce, making it difficult to provide static, relevant few-shot exemplars without a complex retrieval system.
>
> However, we agree that exploring prompt optimization is critical. As detailed in Section 4.4, we employed GEPA to test if best practice prompting could solve the issue. The results revealed a fundamental trade-off rather than a simple fix: enhancing safety against misinformation via prompting currently comes at the cost of making models overly sensitive (on general medical benchmarks), leading them to hallucinate misconceptions in valid queries (on NFP).
>
> > Q1: How was the taxonomy for the seven myth categories derived?
>
> The authors **manually reviewed a diverse subset of the generated questions** to identify recurring thematic patterns in patient misconceptions. We consolidated these themes into six distinct categories and one Others category. Hematology-oncology physicians reviewed these categories to ensure they represented clinically distinct and meaningful types of patient misunderstandings.
>
> > Q2: Can you provide any explanation or additional analysis that gives insight into why performance differs across myth categories?
>
> Because our adversarial questions embed specific myths within complex patient narratives, the difficulty of correcting a false presupposition is inherently tied to the model’s ability to isolate and refute the underlying medical misconception. We observe that performance variations are largely driven by **the depth of specialized clinical expertise required to contradict the premise**.
>
> Qualitatively, categories where models perform relatively well, such as “*No Symptom, No Disease*”, typically involve general medical principles (e.g., the fact that early-stage cancers are frequently asymptomatic). Conversely, low-performing categories like “*Inevitable Side Effect*” often require nuanced, treatment-specific oncology knowledge to refute (e.g., understanding that a permanent colostomy is not inevitable for all colorectal cancer patients). In these specific scenarios, models frequently fail to leverage specialized knowledge, defaulting instead to validating the patient's anxiety rather than correcting the factual error.
>
> We will include this qualitative analysis in the revised discussion section.

---

### Official Review · Reviewer_tQ3u · 2025-11-01

**Soundness:** 2
**Presentation:** 3
**Contribution:** 3
**Rating:** 6
**Confidence:** 4

**Summary:**

The paper targets a clinically important failure mode: when patient questions contain hidden, false assumptions, current LLMs tend to accept the premise rather than correct it. The authors first compare LLMs to human social workers on real patient queries and observe that models are often factually sound yet pragmatically unsafe because they do not challenge misconceptions. They then introduce a physician-verified benchmark of cancer-related questions with embedded false presuppositions, plus a matched control set without them, and show that many strong models still miss these corrections. Finally, they demonstrate that common “precautionary” prompting can
improve myth detection but at the cost of over-caution and degraded performance elsewhere.

**Strengths:**

1. Strong problem framing: focuses on a realistic, safety-critical interaction failure that is underrepresented in exam-style medical benchmarks.
2. Solid evaluation design: includes a no-presupposition control set to quantify over-caution and a category taxonomy that supports actionable error analysis.
3. Good methodological transparency: clear pipeline description and prompt templates; expert involvement to validate that questions actually contain presuppositions and that corrections are medically sound.

**Weaknesses:**

1. Heavy reliance on a single LLM-as-judge for both dataset curation and scoring risks judge-induced bias; human validation is present but limited in scope.
2. The benchmark is largely synthetic and category-balanced, which aids analysis but may distort real-world prevalence and phrasing of patient misconceptions.
3. The initial human comparison uses a small sample and non-physician baselines, which blunts any strong takeaways about relative human vs. LLM performance in clinical settings.

**Questions:**

1. Would you be open to a simple, robust judging setup that combines (i) a small mixture of diverse LLM judges, (ii) a friendly, rubric-driven checklist for presupposition detection/clarification with short evidence spans, and (iii) a tiny human-anchored set to calibrate or weight the ensemble? A short note on rank stability under this setup would make the results feel very solid.
2. Could you add light uncertainty to the main metrics (e.g., straightforward intervals/bootstraps and paired comparisons since models see the same items) and a few compact post-hoc views—such as detection vs. correction breakdown and a couple of counterfactual rewrites—to help readers understand where models stumble?
3. What is your plan to share data, prompts, scoring code, and judge/verifier configurations (including seeds/decoding settings) along with a minimal evaluation harness so others can rerun the scoring or swap in alternative judges? Even a brief “evaluation card” with intended use and caveats would be great.

---

> ### Author Response · Authors · 2025-11-20
>
> We thank the reviewer for their insightful feedback and for recognizing the strong problem framing of our work. We appreciate that you highlighted the solid evaluation design, specifically the inclusion of the NFP set, and the methodological transparency regarding our pipeline and expert verification process. We address your specific questions and concerns below.
>
>
> > W1 & Q1: Would you be open to a simple, robust judging setup that combines (i) a small mixture of diverse LLM judges, (ii) a friendly, rubric-driven checklist…
>
> We believe our current LLM-as-a-judge setup is robust for this specific task for two reasons. First, as detailed in Appendix C.1.1 (Table 2), **LLM-as-a-judge achieves 100% accuracy in binary agreement with humans** on the annotated subset. Second, the verification task is **conceptually easy**. We provide the judge with the specific “ground truth” correction context (as shown in Figure 4 and the prompt in Figure 12). This drastically reduces ambiguity, making the evaluation less about subjective preference and more about factual verification.
>
> We agree that a mixture-of-judges approach is a valuable direction for future work, specifically for more ambiguous medical communication tasks.
>
> > W2: The benchmark is largely synthetic and category-balanced, which aids analysis but may distort real-world prevalence and phrasing of patient misconceptions.
>
> The content of the myths is not synthetic; it is sourced directly from the National Cancer Institute (NCI) and patient advocacy websites, representing the **actual distribution of misconceptions** patients face.
>
> As for phrasing, while the questions are generated, they are **empirically realistic**. As reported in L304–306, in a blinded Turing test against real human questions, the preference for human-written questions was not statistically significant for 80% of the pairs.
>
> > W3: The initial human comparison uses a small sample and non-physician baselines, which blunts any strong takeaways about relative human vs. LLM performance in clinical settings.
>
> We clarify that the pilot study in Section 2 was not intended to make claims on relative human vs. LLM performance in clinical settings. Instead, its primary objective was a **needs assessment** to isolate specific failure modes. While physicians confirmed that frontier LLMs generally provide accurate and helpful medical information, the study served to highlight a critical safety gap, the failure to address false presuppositions, which subsequently motivated the creation of the targeted Cancer-Myth dataset.
>
> > Q2: Could you add light uncertainty to the main metrics...
>
> We have detailed distribution of results in Figure 5, where we can compute the **PCS with 95% bootstrap confidence intervals (CI)** from. As shown in the table below, while the intervals show some overlap among similarly performing models, the ranking tiers remain distinct. We will include this table and a pairwise statistical significance matrix in the revised version.
>
> Models|PCS|95% CI
> :---|:---: |:---:
> GPT-5|0.19|[0.12, 0.25]
> Gemini-2.5-Pro|0.13|[0.06, 0.20]
> Claude-4-Sonnet|0.12|[0.05, 0.18]
> O3|0.05|[-0.02, 0.11]
> Gemini-1.5-Pro|-0.14|[-0.21, -0.08]
> Gemma-2|-0.23|[-0.29, -0.17]
> Claude-3.5-Sonnet|-0.27|[-0.33, -0.20]
> DeepSeek-R1|-0.29|[-0.35, -0.24]
> GPT-4-Turbo|-0.30|[-0.36, -0.25]
> O4-mini|-0.33|[-0.38, -0.27]
> DeepSeek-V3|-0.40|[-0.45, -0.35]
> Qwen-2.5-72B|-0.44|[-0.49, -0.39]
> Qwen-2.5-7B|-0.50|[-0.55, -0.45]
> GPT-4o|-0.52|[-0.57, -0.47]
> Llama-3.1-70B|-0.52|[-0.57, -0.47]
> Llama-4-Scout|-0.59|[-0.63, -0.54]
> Llama-3.1-8B|-0.63|[-0.67, -0.58]
> GPT-3.5|-0.80|[-0.83, -0.76]
> GPT-4o (MDAgents)|-0.84|[-0.87, -0.81]
>
> > Q3: What is your plan to share data, prompts, scoring code…
>
> We will open-source everything.

---

### Official Review · Reviewer_tkx8 · 2025-11-03

**Soundness:** 3
**Presentation:** 2
**Contribution:** 1
**Rating:** 2
**Confidence:** 3

**Summary:**

The authors construct two cancer Q&A datasets, Cancer-Myth, which contains prompts containing misconceptions, and Cancer-Myth-NFP, which contains no false presuppositions. The former contains over 500 prompts while the latter contains just under 200, all LLM-generated. Physicians both verify that prompts contain/do not contain erroneous information, and whether LLM responses correct/do not correct questions.The authors find that even SoTA frontier models cannot correct over 50% of questions containing misconceptions. Adversarial prompts generated by Gemini-1.5-Pro are the most misleading. GEPA optimization improves correction but drops performance on Cancer-NFP and other medical benchmarks.

**Strengths:**

- The datasets appear to be truly challenging, stumping two advanced LLMs. Physician verification helps legitimize this set.
- The prompt generation process is straightforward and can be used with any LLM.
- Well-explained motivation: indeed, providing correct advice without correcting patient misconceptions can be harmful.

**Weaknesses:**

- In my opinion, this is too small of a dataset contribution to be considered useful for the medical LLM research community. It would have made more sense to make both Cancer NFP and Cancer-Myth the same size.
- The GEPA optimization does not seem to be properly tested -- the size of training and test are less than nearly all datasets studied in the GEPA paper.
- 67% identification of questions as human-written versus LLM-generated very clearly indicates that questions appear synthetic overall. This type of distinguishably can be trained on.
- Gemini-2.5-Pro and GPT-5 are not tested as prompt generators, despite being assessed as evaluators.
- The evaluation of NFP is a bit unrealistic. It is possible that a question without presuppositions would never be corrected unless the LLM were asked to engage in such labeling.

**Questions:**

- How were the dataset sizes chosen and why were they not made equal?
- Were Gemini-2.5-Pro and GPT-5 not tested as prompt generators?
- Is there any correlation between being (un)able to correct misconceptions with general performance on other cancer-related Q&A benchmarks?
- How does this approach compare to the DyReMe question generation framework (arxiv 2510.09275)?

---

> ### Author Response · Authors · 2025-11-20
>
> We thank the reviewer for recognizing that Cancer-Myth is a well-motivated, carefully verified, and challenging benchmark that addresses a critical safety issue in medical LLMs. We appreciate the constructive feedback and address the specific concerns below.
>
> > W1 & Q1: Dataset size concerns (585 examples) and unequal splits.
>
> We clarify that **Cancer-Myth is designed as an evaluation benchmark, not a training dataset**.
>  - In the domain of expert-annotated medical QA, our dataset size is comparable to or larger than established evaluation sets. For example, *MedBullets* contains 308 examples and *MedicationQA* contains 690.
>  - Unlike large-scale automated datasets, every example in Cancer-Myth was verified by hematology-oncology physicians. This ensures clinical validity, which is our priority over raw volume.
>  - Regarding the unequal sizes, we started with 888 generated questions, the splits were determined from physicians as detailed in Section 3.3 and L315-317:
>    - **Cancer-Myth (585 examples):** Met criteria for both (1) containing a valid false presupposition and (2) having a medically sound correction.
>    - **Cancer-Myth-NFP (150 examples):** Failed to meet criteria (1). Specifically, these are questions containing no false presupposition.
>    - **Discarded (153 examples):** Met criteria (1) but failed on criteria (2).
>    - **Purpose:** The NFP set is intentionally smaller as it serves as an auxiliary “control group” to measure model over-sensitivity (false positives), whereas the main set measures safety (false negatives).
>
> > W2: GEPA optimization testing and sample size.
>
> GEPA is computationally expensive; In our initial experiments, we observed that 35 examples were sufficient for the prompt optimization to converge. To address the reviewer's concern, we ran a new set of experiments using 140 examples for training/validation (matching the scale used in the original GEPA paper) on Gemini-2.5-Pro. As shown in the table below, increasing the sample size yielded **small performance differences (within 1%)**, confirming that our original findings are robust. We will add this discussion to the revised version of the paper.
>
> Method | Cancer-Myth | Cancer-Myth-NFP | MedQA | PubMedQA | SymCat | Medbullets | Craft-MD
> :---|:---: |:---: |:---: |:---: |:---: |:---: |:---:
> Plain | 41 | 96 | 92 | 82 | 91 | 80 | 68
> GEPA (35 train/val) | 88 | 68 | 85 | 78 | 87 | 72 | 58
> GEPA (140 train/val) | 89 | 68 | 85 | 79 | 86 | 73 | 59
>
> > W3: 67% identification of questions as human-written versus LLM-generated very clearly indicates that questions appear synthetic overall. This type of distinguishably can be trained on.
>
> While Cancer-Myth questions are LLM-generated, empirical evaluation confirms **they closely resemble real patient inquiries**. As noted in L304–306, the preference for human-written questions was not statistically significant for 80% of the paired comparisons, indicating a high degree of realism. On the other hand, while training LLMs to be stylistically indistinguishable from humans is a valid objective, we consider it orthogonal to our primary research goal: testing model safety against critical logical fallacies. Crucially, the “synthetic” nature of the text does not trivialize the medical reasoning challenge. As our results demonstrate, even state-of-the-art models consistently fail to detect these embedded presuppositions.
>
> > W4 & Q2: Gemini-2.5-Pro and GPT-5 are not tested as prompt generators, despite being assessed as evaluators.
>
> The dataset generation and physician annotation were completed **prior to the release of Gemini-2.5-Pro and GPT-5**. However, we argue that this actually strengthens our claims. We show that questions generated by “older” models (like GPT-4o and Gemini-1.5) are sufficient to stump the newer frontier models (Gemini-2.5-Pro and GPT-5). If the newest models cannot solve questions generated by older models, it indicates that the type of difficulty (detecting false presuppositions in patient questions) is a fundamental reasoning hurdle, not a model-specific artifact.

---

> ### Author Response · Authors · 2025-11-20
>
> > W5 & Q3: The evaluation of NFP is a bit unrealistic… Is there any correlation between being (un)able to correct misconceptions with general performance on other cancer-related Q&A benchmarks?
>
> We first clarify the role of the Cancer-Myth-NFP set. NFP is not a standard QA benchmark; it acts as a “negative control” to measure the false positive rate. It tests whether a mitigation strategy simply corrects everything, even valid questions. Our experiments reveal that precautionary prompting strategies like GEPA achieve safety at the cost of precision: while they improve detection on the adversarial set, they cause accuracy on Cancer-Myth-NFP to drop significantly (e.g., from 96% to 68% with Gemini-2.5-Pro), misidentifying valid questions as containing myths. Furthermore, this over-caution imposes a broader penalty, resulting in a 5-15% relative performance drop across standard benchmarks such as MedQA and Medbullets. These findings underscore a critical trade-off in real-world deployment: **enhancing safety against misinformation comes at the cost of making models overly cautious and less accurate on medical queries**.
>
> > Q4: Comparison to DyReMe (Arxiv 2510.09275).
>
> DyReMe (Oct 2025) is concurrent work, released on arXiv after the ICLR deadline. The two studies have distinct scopes: Cancer-Myth specifically isolates the linguistic challenge of false presuppositions to evaluate model safety, whereas DyReMe focuses on creating a general medical benchmark that aligns with real-world patient-doctor communication patterns. We view these efforts as complementary. In fact, DyReMe’s medical data could serve as a valuable seed source to further diversify the communicative styles and scenarios within Cancer-Myth in future iterations.

---

### Author Response · Authors · 2025-12-04
**Summary for AC**

Dear AC,

Across all four reviewers, the feedback is positive about motivation, clinical relevance, expert verification, and the importance of evaluating LLM safety under false-presupposition scenarios. The reviewers primarily raised clarifications rather than fundamental methodological concerns. Below we summarize the key issues and our responses.

1. Dataset size, composition: 585 examples, inclusion criteria (**tkx8**).
- **Already in paper:** Section 3.3 details expert verification and inclusion criteria.
- **Rebuttal:** Size is comparable to other expert-verified medical benchmarks; splits reflect strict physician criteria.

2. GEPA sample size: 35-example GEPA may be too small (**tkx8**).
- **Rebuttal and revision:** 35 examples are sufficient for the prompt optimization to converge; 140-example experiments show <1% difference, confirming robustness.

3. Frontier models (GPT-5 and Gemini-2.5-Pro) not used as generators (**tkx8**).
- **Rebuttal:** Dataset predates these releases; older-model questions still stump newer models, indicating a fundamental challenge.

4. Role of NFP set (**tkx8**).
- **Already in paper:** Measure the false positive rate. Section 4.4, safety improvements (GEPA) reduce misinformation but over-correct valid queries on NFP set (e.g., 96% → 68%) and drop performance 5–15% on general medical benchmarks, highlighting a real-world trade-off.

5. LLM-as-judge protocol not good enough (**tQ3u**).
- **Already in paper:** Appendix C.1.1 shows judge consistency.

6. Synthetic vs. real misconceptions: Generated phrasing or category balance may distort realism (**tQ3u**).
- **Already in paper:** Section 3.1, myths sourced from NCI/patient-advocacy sites. Only phrasing is generated; content distribution reflects real-world misconceptions.

7. Category-level performance differences (**ct9K**).
- **Rebuttal and revision:** Variation reflects clinical specialization required; low-performing categories need nuanced oncology knowledge.

8. Thinking models / test-time compute (**1HfB**).
- **Already in paper:** In Figure 5, GPT-5, Gemini-2.5-Pro, and Claude-4-Sonnet, DeepSeek-R1 are thinking models.
- **Rebuttal and revision:** New experiments with o3 and o4-mini; increased test-time compute does not clearly reduce the tendency to accept false premises.

9. Expert-in-the-loop protocol transparency (**1HfB**).
- **Already in paper:** L293–300, Appendix C.1.3 detail experts, criteria, agreement, and adjudication.
- **Rebuttal and revision:** Added a data statement in Appendix B.

10. Statistical uncertainty (**tQ3u**).
- **Rebuttal and revision:** 95% bootstrap CIs computed; ranking tiers unchanged.

---

### Meta-Review · Area_Chair_ruz8 · 2026-01-06

**Summary:**

All the reviews are positive about the importance of the problem (LLMs failing to correct false presuppositions in patient questions) and credible dataset construction. This is extremely important in teh times when general public use AI as a tool for clinical support. The reviewers felt that the data set was constructed carefully, praised the framing and actionable metrics.

Key concerns of the reviewers include the data set size, the limitations of the synthetic datasets, LLM-as-judge-risks and the lack of transparancy in expert reviews.

**Reviewer Concerns:**

Some of the concerns are addressed in the rebuttal. Details of the inclusion and exclusion criteria, the sample size, the frontier models, the peroformance differences etc.

Some remain -- the justification fo the data set size, category-level performance difference understanding, transparancy (moved to appendix when it is important).

**Reviewer Scores:**

I can only see reviewer tkx8 improving the score to a 4 or so. But I still see that some of the reviews would not be fully satisfied. There is one reviewer who is a clear champion (although I feel that the score is over inflated).

---

### Decision · Program_Chairs · 2026-01-26

Accept (Poster)